# Analytical Determination of the Nucleation-Prone, Low-Density Fraction of Subcooled Water

**DOI:** 10.3390/e22090933

**Published:** 2020-08-25

**Authors:** Olaf Hellmuth, Rainer Feistel

**Affiliations:** 1Leibniz Institute for Tropospheric Research (TROPOS), Permoserstraße 15, D-04318 Leipzig, Germany; 2Leibniz Institute for Baltic Research (IOW), Seestraße 15, D-18119 Rostock-Warnemünde, Germany; rainer.feistel@io-warnemuende.de

**Keywords:** IAPWS G12-15, equation of state of subcooled water, low-density and high-density water fraction

## Abstract

Subcooled water is the primordial matrix for ice embryo formation by homogeneous and heterogeneous nucleation. The knowledge of the specific Gibbs free energy and other thermodynamic quantities of subcooled water is one of the basic prerequisites of the theoretical analysis of ice crystallization in terms of classical nucleation theory. The most advanced equation of state of subcooled water is the IAPWS G12-15 formulation. The determination of the thermodynamic quantities of subcooled water on the basis of this equation of state requires the iterative determination of the fraction of low-density water in the two-state mixture of low-density and high-density subcooled water from a transcendental equation. For applications such as microscopic nucleation simulation models requiring highly frequent calls of the IAPWS G12-15 calculus, a new two-step predictor-corrector method for the approximative determination of the low-density water fraction has been developed. The new solution method allows a sufficiently accurate determination of the specific Gibbs energy and of all other thermodynamic quantities of subcooled water at given pressure and temperature, such as specific volume and mass density, specific entropy, isothermal compressibility, thermal expansion coefficient, specific isobaric and isochoric heat capacities, and speed of sound. The misfit of this new approximate analytical solution against the exact numerical solution was demonstrated to be smaller than or equal to the misprediction of the original IAPWS G12-15 formulation with respect to experimental values.

## 1. Introduction

The knowledge of the properties of supercooled or subcooled water (the notion used here) is of outstanding importance for a variety of natural and technical processes related to phase transitions of water such as the microphysical evolution of atmospheric clouds (see, e.g., Meyers et al. [1], Khvorostyanov and Sassen [2], Lohmann and Kärcher [3], Lohmann et al. [4], Pruppacher and Klett [5], Heymsfield et al. [6], Jensen and Ackerman [7], Barahona and Nenes [8], Jensen et al. [9], Zasetsky et al. [10], Khvorostyanov and Curry [11], Khvorostyanov and Curry [12], Hellmuth et al. [13], Khvorostyanov and Curry [14], and Lohmann et al. [15]), the cryopreservation of organelles, cells, tissues, extracellular matrices, organs, and foods (see, e.g., Pegg [16] and Espinosa et al. [17,18]); and water vitrification (see, e.g., Debenedetti and Stanley [19], Bhat et al. [20], and Zobrist et al. [21]). Subcooled water is the primordial matrix for ice embryo formation by homogeneous and heterogeneous nucleation. Recently, Hellmuth et al. [22] studied the variation of the homogeneous freezing rate of subcooled water in dependence on pressure and temperature. Within the framework of classical nucleation theory both the thermodynamic driving force of freezing and the surface tension of the water–ice system were scaled in terms of fundamental thermodynamic properties of the macrophases of subcooled water and hexagonal ice, such as the Gibbs energy, entropy, enthalpy, mass density, isobaric heat capacity, isothermal compressibility, and isobaric thermal expansivity. These quantities appeared the generating ones for the determination of the nucleation metrics.

For applications to the aforementioned questions of interest, several equations of state (EoS) for subcooled water became available (for a brief review see Holten et al. [23]). In their study on ice nucleation, Hellmuth et al. [22] employed the Thermodynamic Equation Of Seawater 2010 (TEOS-10), which is an international standard for the thermodynamic properties of seawater, ice, and humid air, adopted in 2009 by the Intergovernmental Oceanographic Commission (IOC) of the UNESCO and in 2011 by the International Union of Geodesy and Geophysics (IUGG) (Feistel et al. [24,25]; Wright et al. [26]; IOC, SCOR, and IAPSO [27]; Feistel [28]; Feistel [29]; TEOS-10 website http://www.teos-10.org/). This seawater standard is based on thermodynamic potentials for fluid water, hexagonal ice, seasalt dissolved in water, and dry air. The fluid water part is described according to Wagner and Pruß [30] and IAPWS R6-95 [31], known as “IAPWS-95”, and took into account quality-assured experimental data which were available by the middle of the year 1994. For details of TEOS-10 and IAPWS-95 the reader is referred to the Appendix A.

In order to better represent the thermodynamic properties of subcooled liquid water, Holten et al. [23,32,33,34] and Holten and Anisimov [35] proposed a new EoS of subcooled water, which considers the experimental data of liquid water in the metastable region of the phase diagram above the homogeneous nucleation temperature. The thermodynamic formulation of this EoS is a mean-field version of an EoS developed by Holten and Anisimov [35]. It is based on the assumption that liquid water is a mixture of a high-density structure A and a low-density structure B of water (“two-state model”, cf. Section 3.1). Such an assumption is justified by empirical findings (see references in Holten and Anisimov [35]). This new EoS was implemented in the IAPWS guideline IAPWS G12-15 [36], given in form of the dimensionless Gibbs energy of subcooled water as a function of temperature, pressure, and the equilibrium mole fraction of the low-density structure of water in the two-state mixture. Having at ones disposal the Gibbs energy of subcooled water, one can determine the mass density, the specific entropy, the isothermal compressibility, the isothermal expansivity, the specific isobaric and isochoric heat capacities, and the speed of sound of subcooled water. While temperature and pressure serve as the independent variables, the equilibrium mole fraction of low-density water is a dependent variable, which enters the determination of the Gibbs energy in form of a function of pressure and temperature. The determination of the low-density water mole fraction requires the numerical solution of a transcendental equation, which serves as a physical constraint for the water composition at prescribed pressure and temperature.

For applications which require frequent calls of the EoS such as for computational fluid dynamics (CFD) modeling, the iterative determination of the low-density water mole fraction might be too time-consuming and should be circumvented. Alternatively, one can either precalculate look-up tables of the low-density water mole fraction with sufficiently high resolution in pressure and temperature, from which the sought after value can be determined by interpolation in two dimensions, or one can try to find a sufficiently accurate approximative solution, which avoids the numerical solution of a transcendental equation. In the present study, we will follow the second attempt.

The aim of the present study is twofold: (i) derivation of an approximative solution of the transcendental equation for the low-density water mole fraction, which allows a highly accurate analytical determination of this quantity as a function of pressure and temperature, and (ii) determination of the deviations of the resulting approximative values of the aforementioned thermodynamic variables from their numerical (“exact”) counterparts throughout the validity range of the underlying EoS. The paper is organized as follows. In Section 2, the material and the method of the present analysis is briefly described. Section 3 addresses the theoretical background of the analysis. In Section 4, the new calculus is applied to the IAPWS G12-15 [36] formulation for subcooled water, and its deviation from the traditional numerical method is analyzed. Section 5 contains the discussion of the results. Finally, the conclusions are drawn in Section 6. The paper is completed by Appendix B containing the algorithm of the calculus in a compact form and by Appendix A, which includes additional explanations, details of the calculus, and a comprehensive table part displaying the verification results. The enumeration of sections, equations, and tables which are presented in the Appendix A, is specified with the prefix SM.

## 2. Materials and Methods

The starting point of the present analysis is the IAPWS G12-15 [36] formulation for subcooled water, the calculus of which is presented in Section SM-3. The subject of interest of the present study is the governing equation for the mole fraction of low-density water therein, given by Equation (SM-3.8), which is a transcendental equation and ordinarily requires a numerical iteration to find the roots. However, by means of a Taylor expansion up to the quadratic term (quadratization), this equation is brought into the form of a cubic equation, which in turn can be solved analytically by means of Cardano’s method presented in the Appendix A. To define a benchmark for the subsequent verification of the new analytical solution, at first a numerical solver was implemented and evaluated (cf. Section SM-5). On this basis the differences between the traditional numerical and the new analytical solution of the mole fraction of subcooled water together with the thermodynamic deliveries of the IAPWS G12-15 [36] formulation are determined as functions of pressure and temperature, such as the specific Gibbs energy, the specific volume and mass density, the specific entropy, the isothermal compressibility, the thermal expansion coefficient, the specific isobaric and isochoric heat capacities, and speed of sound (cf. Sections SM-6 and SM-7). In order to assess the goodness of the analytical solution of the EoS, the misprediction of the analytical form with respect to the observations is compared with those of the numerical form. For the comparison, the experimental data of Holten et al. [23] are used.

## 3. Theoretical Background

### 3.1. On the Two-State Character of Subcooled Water

The observed anomalous behavior of the compressibility, the thermal expansivity, and viscosity of water and the existence of mass density maximum already prompted Röntgen [37] to view subcooled water as a two-state molecular mixture. According to Röntgen’s perception, molecules of the first kind are ice molecules, which go over into molecules of the second kind upon heat supply. If in turn heat is extracted from the liquid, then a corresponding part of ice molecules will reappear. Subcooled water is considered a saturated solution of ice molecules; the saturated fraction of which depends on temperature. The saturated concentration of ice molecules increases with decreasing temperature. Röntgen [37] further assumed that the transition of molecules of the first kind into the second kind is accompanied by a decrease of the volume of the mixture, as can be observed during melting of ice. In its modern form, this perception is the basis for the IAPWS G12-15 [36] EoS, the rationale of which will be briefly outlined hereafter. According to Holten et al. [34], subcooled water is assumed to be a mixture of two interconvertible states or structures of water: a high-density state A and a low-density state B. The fraction of molecules in state B is denoted by *x*, and is controlled by the following equilibrium pseudo-reaction (Holten et al. [34], Equation (Equation 1) therein): (1)A⇌B.

As argued by those authors, the states A and B could correspond to different arrangements of the hydrogen-bonded network or to two kinds of local coordination of the water molecules. The molar Gibbs energy of a two-state mixture reads (Holten et al. [34], Equation (Equation 2) therein):(2)$$G=G^{A}+xG^{\mathrm{BA}}+RT\left[x\ln x+(1-x)\ln(1-x)\right]+G^{E}\;.$$
Here, *R* denotes the universal gas constant, *T* is the temperature, $G^{A}$ is the molar Gibbs energy of the pure structure A, defines the background of the properties, and can be approximated by a well-defined polynomial as a function of *T* and *p*, i.e., $G^{A}=G^{A}\left(T,p\right)$. The quantity $G^{\mathrm{BA}}\equiv G^{B}-G^{A}$ is the difference in the molar Gibbs energies between the pure configurations A and B, and $G^{E}$ is the excess Gibbs energy of mixing. The difference $G^{\mathrm{BA}}$ is related to the equilibrium constant K(T,p) of the pseudo-reaction given by Equation (Equation 1), which reads (Holten et al. [34], Equation (Equation 3) therein):(3)$$\ln K\left(T,p\right)=\frac{G^{\mathrm{BA}}}{RT}\;.$$

From Equation (Equation 3), it follows that $G^{\mathrm{BA}}=G^{\mathrm{BA}}\left(T,p\right)$. The excess Gibbs energy $G^{E}$ originates from the non-ideality of the mixture and is given by the sum of contributions of the enthalpy of mixing $H^{E}$ and of the excess entropy $S^{E}$ (Holten et al. [34], Equation (Equation 6) therein):(4)$$G^{E}=H^{E}-TS^{E}\;.$$

Subcooled water is approximated by an athermal solution, i.e., the molecules of the two liquids are allowed to differ appreciably in size, and strong orienting forces caused by dipoles or hydrogen bonding are allowed to occur. In such case, $H^{E}=0$ and $S^{E}=-R\omega x\left(1-x\right)$, where ω(p) denotes the pressure-dependent molecular interaction parameter, which accounts for the non-ideality of the solution, resulting in $G^{E}=G^{E}\left(x,T,p\right)$. Knowing *x*, the EoS of subcooled water delivers the molar Gibbs energies of the pure configurations A and B, $G^{A}\left(T,p\right)$ and $G^{B}\left(T,p\right)$, as well as that of the mixture, G(x,T,p).

The two-state character of subcooled water has implications for ice crystallization which arise from an empirical judgment of Ostwald [38] (pp. 306–309 therein), which has later become known as Ostwald’s rule of stages. The implications of this rule are discussed in Section SM-2.

### 3.2. Analytical Determination of the Fraction of Low-Density Water in the Two-State Water Mixture

For the sake of completeness, the full calculus of the IAPWS G12-15 [36] formulation of the properties of subcooled water is presented in Section SM-3. From this section we excerpt here only those parts of the calculus which are directly related to the determination of the low-density water mole fraction.

Introducing the reduced temperature τ and the reduced pressure π (cf. Section SM-3.1, Equation (SM-3.2)):(5)$$\tau =\frac{T}{T_{\mathrm{LL}}}-1\;,\;\pi =\frac{p}{{\hat{\varrho }}_{0}R_{W}T_{\mathrm{LL}}}\;,$$
with $T_{\mathrm{LL}}$, ${\hat{\varrho }}_{0}$, and $R_{W}$, respectively, denoting the temperature of the liquid-liquid critical point, the reference mass density of subcooled water, and the specific gas constant of water vapor, the equilibrium mole fraction of low-density water, $x_{e}$, was demonstrated to obey the following transcendental equation (cf. Section SM-3.1, Equation (SM-3.8)):(6)$$F\left(x_{e}\right)=L\left(\tau ,\pi \right)+\ln\left(\frac{x_{e}}{1-x_{e}}\right)+\omega \left(\pi \right)\left(1-2x_{e}\right)=0\;.$$

The dimensionless quantities L(τ,π) and ω(π) denote the ordering field and the interaction parameter, respectively, and are well-defined functions of τ and π (cf. Section SM-3.1, Equations (SM-3.5) and (SM-3.6)). The mixed state of subcooled water is characterized by the condition $0<x_{e}<1$. Knowing $x_{e}\;=\;x_{e}\left(\tau ,\pi \right)$ from the numerical solution of Equation (Equation 6), the dimensionless Gibbs free energy ψ(τ,π) is fully determined and given in analytical form, from which all other thermodynamic parameters can be analytically determined (cf. Section SM 3.2).

To derive an approximate analytical solution of Equation (Equation 6) we introduce the auxiliary parameters
(7)α=2ω,β=−(L+ω),
by means of which Equation (Equation 6) can be rearranged as follows (the subscript “e” for the equilibrium value of *x* will be omitted hereafter):(8)$$\ln\left(\frac{x}{1-x}\right)=\alpha x+\beta \;⇝\;\frac{x}{1-x}=e^{\beta }e^{\alpha x}\;.$$

In order to obtain a first approximation of the mole fraction of low-density water, $x\approx \bar{x}$, the following quadratization by expansion of the exponential term in a Taylor series will be employed:(9)$$e^{\alpha \bar{x}}\approx 1+\alpha \bar{x}+\frac{1}{2}\alpha^{2}{\bar{x}}^{2}\;,\;\bar{x}\ll 1\;.$$

Therewith, Equation (Equation 8) can be rearranged into a cubic equation for the determination of $\bar{x}$:(10)$$\begin{array}{c} \bar{A}\;{\bar{x}}^{3}+\bar{B}\;{\bar{x}}^{2}+\bar{C}\;\bar{x}+\bar{D}=0\;,\;\\ \bar{A}=-\frac{\alpha^{2}}{2}e^{\beta }=-\frac{\alpha^{2}}{2}\bar{D}\;,\;\\ \bar{B}=\alpha \left(\frac{\alpha }{2}-1\right)e^{\beta }=\alpha \left(\frac{\alpha }{2}-1\right)\bar{D}\;,\;\\ \bar{C}=e^{\beta }\left(\alpha -1\right)-1=\bar{D}\left(\alpha -1\right)-1\;,\;\\ \bar{D}=e^{\beta }\;. \end{array}$$

The parameters $\bar{A}$, $\bar{B}$, $\bar{C}$, and $\bar{D}$ are well-defined functions of τ and π. Equation (Equation 10) can be solved using Cardano’s method and has three real and/or complex solutions, ${\bar{x}}_{i}={\bar{x}}_{i}\left(\bar{A},\bar{B},\bar{C},\bar{D}\right)$, i=1,2,3, representing the first guess of the mole fraction of low-density water. We are only interested in the real solution of Equation (Equation 10), which reads (cf. Section SM-4, Equation (SM-4.10))
(11)$$\begin{array}{ccc} \bar{x} & = & \sqrt[{3}]{-\frac{\bar{q}}{2}+\sqrt{\bar{\Delta }}}+\sqrt[{3}]{-\frac{\bar{q}}{2}-\sqrt{\bar{\Delta }}}\;,\;\\ \bar{\Delta } & = & {\left(\frac{\bar{q}}{2}\right)}^{2}+{\left(\frac{\bar{p}}{3}\right)}^{3}\geq 0\;,\;\\ \bar{p} & = & \bar{s}-\frac{{\bar{r}}^{2}}{3}\;,\;\bar{q}=\frac{2{\bar{r}}^{3}}{27}-\frac{\bar{s}\;\bar{r}}{3}+\bar{t}\;,\;\\ \bar{r} & = & \frac{\bar{B}}{\bar{A}}\;,\;\bar{s}=\frac{\bar{C}}{\bar{A}}\;,\;\bar{t}=\frac{\bar{D}}{\bar{A}}\;. \end{array}$$

A prerequisite for the existence of a real solution in the form given by Equation (Equation 11) is the non-negativity of the radicand $\bar{\Delta }$, the fulfillment of which is ensured throughout the p−T range analyzed here.

To obtain the second guess of *x*, the mole fraction of low-density water is decomposed into the first guess $\bar{x}$ and an increment $x^{\prime }$:(12)$$x≅\bar{x}+x^{\prime }\;.$$

Inserting Equation (Equation 12) into Equation (Equation 8) one obtains
(13)$$\frac{\bar{x}+x^{\prime }}{1-\bar{x}-x^{\prime }}=e^{\alpha \bar{x}+\beta }e^{\alpha x^{\prime }}\;.$$

Employing again the quadratization
(14)$$e^{\alpha x^{\prime }}\approx 1+\alpha x^{\prime }+\frac{1}{2}\alpha^{2}x^{\prime 2}\;,\;x^{\prime }\ll 1\;,$$

Equation (Equation 13) can be rearranged into a cubic equation for the determination of $x^{\prime }$:(15)$$\begin{array}{c} A^{\prime }x^{\prime 3}+B^{\prime }x^{\prime 2}+C^{\prime }x^{\prime }+D^{\prime }=0\;,\;\\ A^{\prime }=-\frac{\alpha^{2}}{2}e^{\alpha \bar{x}+\beta }=\bar{A}e^{\alpha \bar{x}}\;,\;\\ B^{\prime }=\alpha \left[\frac{\alpha }{2}\left(1-\bar{x}\right)-1\right]e^{\alpha \bar{x}+\beta }=A^{\prime }\left(\frac{\bar{B}}{\bar{A}}+\frac{1}{\alpha }\ln\frac{A^{\prime }}{\bar{A}}\right)\;,\;\\ C^{\prime }=\left[\alpha (1-\bar{x})-1\right]e^{\alpha \bar{x}+\beta }-1=\frac{A^{\prime }}{\bar{A}}\left(1+\bar{C}-\bar{D}\ln\frac{A^{\prime }}{\bar{A}}\right)-1\;,\;\\ D^{\prime }=\left(1-\bar{x}\right)e^{\alpha \bar{x}+\beta }-\bar{x}=-\frac{1}{\alpha }\left(1+\frac{\bar{D}A^{\prime }}{\bar{A}}\right)\ln\frac{A^{\prime }}{\bar{A}}+\frac{\bar{D}A^{\prime }}{\bar{A}}\;. \end{array}$$

The parameters $A^{\prime }$, $B^{\prime }$, $C^{\prime }$, and $D^{\prime }$ are well-defined functions of τ and π. Equation (Equation 15) is of the same structure as Equation (Equation 10) and has three real and/or complex solutions, $x_{i}^{\prime }=x_{i}^{\prime }\left(A^{\prime },B^{\prime },C^{\prime },D^{\prime }\right)$, i=1,2,3, representing the residual value of the low-density water mole fraction. Again, we are only interested in the real solution of Equation (Equation 15), which reads (cf. Section SM-4, Equation (SM-4.10)):(16)$$\begin{array}{ccc} x^{\prime } & = & \sqrt[{3}]{-\frac{q^{\prime }}{2}+\sqrt{\Delta^{\prime }}}+\sqrt[{3}]{-\frac{q^{\prime }}{2}-\sqrt{\Delta^{\prime }}}\;,\;\\ \Delta^{\prime } & = & {\left(\frac{q^{\prime }}{2}\right)}^{2}+{\left(\frac{p^{\prime }}{3}\right)}^{3}\geq 0\;,\;\\ p^{\prime } & = & s^{\prime }-\frac{r^{\prime 2}}{3}\;,\;q^{\prime }=\frac{2r^{\prime 3}}{27}-\frac{s^{\prime }\;r^{\prime }}{3}+t^{\prime }\;,\;\\ r^{\prime } & = & \frac{B^{\prime }}{A^{\prime }}\;,\;s^{\prime }=\frac{C^{\prime }}{A^{\prime }}\;,\;t^{\prime }=\frac{D^{\prime }}{A^{\prime }}\;. \end{array}$$

A prerequisite for the existence of a real solution in the form given by Equation (Equation 16) is the non-negativity of the radicand $\Delta^{\prime }$, the fulfillment of which is also ensured throughout the p−T range analyzed here.

Knowing the first guess and the incremental value of the low-density water mole fraction, $\bar{x}$ and $x^{\prime }$, the second guess of *x* is given by Equation (Equation 12). The analytical solution given by this equation can be considered as a two-step predictor-corrector approximation.

### 3.3. Analyses of the Pure States of Water (Limiting Cases)

The limiting cases of unary water, i.e., of water consisting either of pure low-density water with mole fraction *x* = 1 or of pure high-density water with mole fraction *x* = 0, set constraints on the ordering field L(τ,π) and on the interaction parameter ω(π). In order to determine whether these constraints can be achieved in the definition range of the EoS, we start our consideration with Equations (Equation 7) and (Equation 8):(17)$$f\left(x\right)=\ln\left(\frac{x}{1-x}\right)=\alpha x+\beta \;,\;\alpha =2\omega \;,\;\beta =-\left(L+\omega \right)\;.$$
The transition to the high-density state of water requires the fulfillment of the following constraint:
(18)$$\lim_{x\to 0}f\left(x\right)=-\left(L+\omega \right)=-\infty \;.$$Because of the finiteness of the interaction parameter, which is restricted to the interval $2\leq \omega \left(\pi \right)\leq \omega_{\max}\left(\pi_{\max}\right)\ll \infty$, the fulfillment of the constraint given by Equation (Equation 18) requires the ordering field approaching L(τ,π)→∞.Analogously, the transition to the low-density state of water requires the fulfillment of the following constraint:
(19)$$\lim_{x\to 1}f\left(x\right)=-\left(L-\omega \right)=\infty \;.$$Because of the restriction of the interaction parameter to the interval $2\leq \omega \left(\pi \right)\leq \omega_{\max}\left(\pi_{\max}\right)\ll \infty$, the fulfillment of the constraint given by Equation (Equation 19) requires the ordering field approaching L(τ,π)→−∞.

Figure 1 displays the ordering field L(τ,π) according to the Appendix A in the definition range 0≤τ≤10 and 0≤π≤10 imbedding the validity subrange of the EoS.

It can be seen that L(τ,π) is a positive definite and finite quantity (no singularities) in the EoS validity range. As a consequence, the pure states of either low-density or high-density water corresponding to x=0 or x=1 are thermodynamically not accessible in the validity range of the EoS.

## 4. Application to the IAPWS G12-15 Formulation

### 4.1. Computer Program Verification

The IAPWS G12-15 [36] formulation, as described in Section SM-3, serves as a reference calculus. To ensure the correct implementation of the calculus, we have compared our results with the table values provided by IAPWS G12-15 [36] (Table 5 therein) for computer program verification, presented here in Section SM-3.5, Table SM-3.5.1. Equation (Equation 6) was numerically solved with consideration of the constraints on the subintervals for the numerical determination of the equilibrium mole fraction of low-density water, $x_{e}$, specified in Section SM-3.4.

For the numerical root finding of the transcendental Equation (Equation 6) we used the standard FORTRAN routine SUBROUTINE
zbrak(fx,x1,x2,n,xb1,xb2,nb) from Press et al. [39] (Section 9.1 therein) for root bracketing employing the intermediate value theorem. The quantity fx(x) is given by the function $F(x_{e})$ defined by Equation (Equation 6) on the interval from x1 to x2 which is subdivided in n equally spaced segments. The root-finding algorithm searches for the zero crossings of the function fx(x) = 0. The integer nb is an input variable and specifies the maximum number of roots sought, and is reset to the number of bracketing pairs xb1(1:nb), xb2(1:nb) that are found. Depending on the parameters ω(π) and L(τ,π), the search for the root was executed separately in the three different intervals [x1,x2] defined in Section SM-3.4, Table SM-3.4.1. Each of these three search intervals was subdivided into n subintervals. In the employed standard routine, the variable n is declared as of type INTEGER with an allowed maximum value of 2.147.483.647. In the present case, we set n = $10^{9}$ for the reference solution, allowing $x_{e}$ to be accurate to $10^{-9}$.

Section SM-5, Table SM-5.1 shows the deviation of the numerically determined mass density $\hat{\varrho }$, the thermal expansion coefficient $\alpha_{p}$, the isothermal compressibility $\kappa_{T}$, the isobaric heat capacity $c_{p}$, the speed of sound *w*, the equilibrium mole fraction of low-density water $x_{e}$, and the ordering field *L* using the numerical root finder SUBROUTINE
zbrak(fx,x1,x2,n,xb1,xb2,nb) with n = $10^{9}$ from the corresponding IAPWS G12-15 [36] reference values (subscript 🟉) provided in Section SM-3.5, Table SM-3.5.1. Except for the signed quantity $\alpha_{p}$ all differences are given as relative deviations in units of parts per billion (ppb). The absolute values of the relative deviations for the equilibrium mole fraction of low-density water, $x_{e}$, and for the mass density, $\hat{\varrho }$, are rather small (<4ppb). The quantity *L* (last column) is a function of τ and π only but not of $x_{e}$, i.e., the recognized relative deviation of *L* from its references value originates from the specific way of implementation. Note, to reproduce the values in Section SM-3.5, Table SM-3.5.1 to the number of digits given, the solution for the fraction $x_{e}$ should be accurate to $10^{-10}$. However, the recognized deviations from the reference values are extremely small and originate from error propagation as a consequence of the somewhat lower accuracy of the calculated $x_{e}$ values (employing n =
$10^{9}$). Therefore, the implementation of the IAPWS G12-15 [36] formulation is correct. For comparison, Table SM-5.1 also displays the results for n =$10^{k}$, k = 8,…,1. In general, upon decreasing n the deviations remarkably increase. For n=10 the relative deviation of $x_{e}$ increases to ≈21%, and of $\hat{\varrho }$ to ≈1.2% at atmospheric pressure. On the base of the results presented in Section SM-5, Table SM-5.1, one can state the correct implementation of the numerical solver of the IAPWS G12-15 [36] formulation (for benchmark).

### 4.2. Deviations of the Analytical from the IAPWS G12-15 Reference Formulation

Section SM-6, Table SM-6.1 displays the same type of information as Section SM-5, Table SM-5.1 but now referring to the deviation of the analytically calculated thermodynamic values from the corresponding IAPWS G12-15 [36] reference values shown in Section SM-3, Table SM-3.5.1. The presented results are for n =$10^{9}$. The analysis revealed the fulfillment of the constraints $\bar{\Delta }$≥ 0 and $\Delta^{\prime }$≥ 0 in the T−p range, i.e., the analytical solution is given by Equation (Equation 12) with Cardano’s solutions (Equation 11) and (Equation 16). The deviations were found to be rather small except for deeply subcooled water at *T* = 235.15K and *p* = 0.101325MPa, where the largest deviation from the IAPWS G12-15 [36] reference values occurs. At these conditions the absolute values of the relative deviations amount ≈1.6‰ for the low-density water fraction, ≈2.6‰ for the isothermal compressibility, ≈2.3‰ for the isobaric heat capacity, and ≈0.06‰ for the speed of sound, respectively. The absolute deviation for thermal expansion coefficient amounts $0.15\cdot 10^{-4}\;K^{-1}$ at T=225.15K, resulting in a relative deviation of ≈5.1‰ by virtue of a thermal expansion coefficient of $\approx -29.634\cdot 10^{-4}\;K^{-1}$ (cf. Section SM-3.5, Table SM-3.5.1).

### 4.3. Performance of the Analytical form of the EoS

In the following, the IAPWS G12-15 [36] formulation with use of the root finder SUBROUTINE
zbrak(fx,x1,x2,n,xb1,xb2,nb) for the determination of the low-density water fraction will be called “numerical form of the EoS” (thermodynamic properties subscripted with “num”), and the IAPWS G12-15 [36] formulation using the analytical solution for $x_{e}$ presented in Section 3.2 will be called “analytical form of the EoS” (values without subscript). The results are presented in form of table values for the deviations of the analytical form from the numerical form in different ranges of *p* and *T*. These ranges were chosen according to the availability of experimental data of the different thermodynamic quantities predicted by the IAPWS G12-15 [36] formulation. A comprehensive comparison of the IAPWS G12-15 [36] predictions with experimental data has been performed previously by Holten et al. [23]. The aim of the present study is to assess the accuracy of the analytical form by intercomparison of the differences between the analytical and numerical forms of the EoS with the differences between the numerical form and the experimental data analyzed by Holten et al. [23]. If the absolute value of the former difference is lower than or equal to the absolute value of the latter one, then the approximative form can be considered acceptable.

The results of the analysis on the performance of the approximative form are presented in Section SM-7 and comprise the following tables.
Section SM-7.1: Mass density
-Table SM-7.1.1: Relative deviation, $\left(\hat{\varrho }-{\hat{\varrho }}_{\mathrm{num}}\right)/{\hat{\varrho }}_{\mathrm{num}}$ in parts per billion (ppb), of the mass density $\hat{\varrho }$ using the analytically determined low-density water fraction from the mass density ${\hat{\varrho }}_{\mathrm{num}}$ using the numerically determined low-density water fraction at *p* = 0.101325MPa.-Table SM-7.1.2: As Table SM-7.1.1, but for 0.1≤ p/MPa≤400 and 253.15≤T/K≤303.15.-Table SM-7.1.3: As Table SM-7.1.1, but for 0.1≤p/MPa≤1000 and 235.15≤T/K≤303.15.-Table SM-7.1.4: As Table SM-7.1.1, but for 200≤p/hPa≤1000 and 235.15≤T/K≤300.15.Section SM-7.2: Thermal expansivity
-Table SM-7.2.1: Deviation, $\left(\alpha_{p}-\alpha_{p,\mathrm{num}}\right)/\left(10^{-4}\;K^{-1}\right)$, of the thermal expansivity $\alpha_{p}$ using the analytically determined low-density water fraction from the thermal expansivity $\alpha_{p,\mathrm{num}}$ using the numerically determined low-density water fraction for 0.1≤p/MPa≤600 and 245.5≤T/K≤288.0.Section SM-7.3: Isothermal compressibility
-Table SM-7.3.1: Relative deviation, $\left(\kappa_{T}-\kappa_{T,\mathrm{num}}\right)/\kappa_{T,\mathrm{num}}$ in parts per billion (ppb), of the isothermal compressibility $\kappa_{T}$ using the analytically determined low-density water fraction from the thermal expansivity $\kappa_{T,\mathrm{num}}$ using the numerically determined low-density water fraction for 0.1≤p/MPa≤190 and 235.15≤T/K≤300.15.Section SM-7.4: Isobaric heat capacity
-Table SM-7.4.1: Relative deviation, $\left({\hat{c}}_{p}-{\hat{c}}_{p,\mathrm{num}}\right)/{\hat{c}}_{p,\mathrm{num}}$ in parts per billion (ppb), of the isobaric heat capacity ${\hat{c}}_{p}$ using the analytically determined low-density water fraction from the isobaric heat capacity ${\hat{c}}_{p,\mathrm{num}}$ using the numerically determined low-density water fraction for 0.1≤p/MPa≤190 and 235.15≤T/K≤300.15.Table SM-7.4.2: As Table SM-7.4.1, but for 200≤p/hPa≤1000 and 235.15≤T/K≤300.15.Section SM-7.5: Sound speed
-Table SM-7.5.1: Relative deviation, $\left(w-w_{\mathrm{num}}\right)/w_{\mathrm{num}}$ in parts per billion (ppb), of the sound speed *w* using the analytically determined low-density water fraction from the sound speed $w_{\mathrm{num}}$ using the numerically determined low-density water fraction for 0.1≤p/MPa≤400 and 253.15≤T/K≤303.15.-Table SM-7.5.2: As in Table SM-7.5.1, but for 0.1≤p/MPa≤1000 and 273.15≤T/K≤303.15.Section SM-7.6: Gibbs energy and entropy
-Table SM-7.6.1: Relative deviation, $\left(\hat{g}-{\hat{g}}_{\mathrm{num}}\right)/{\hat{g}}_{\mathrm{num}}$ in units of parts per billion (ppb), of the specific Gibbs energy $\hat{g}$ using the analytically determined low-density water fraction (Section 3.2) from the specific Gibbs energy ${\hat{g}}_{\mathrm{num}}$ using the numerically determined low-density water fraction for 200≤p/hPa≤1000 and 235.15≤T/K≤300.15.-Table SM-7.6.2: Relative deviation, $\left(\hat{s}-{\hat{s}}_{\mathrm{num}}\right)/{\hat{s}}_{\mathrm{num}}$ in parts per billion (ppb), of the specific entropy $\hat{s}$ using the analytically determined low-density water fraction from the specific entropy ${\hat{s}}_{\mathrm{num}}$ using the numerically determined low-density water fraction for 200≤p/hPa≤1000 and 235.15≤T/K≤300.15.

## 5. Discussion

Table 1 displays the summary of the evaluation of the table values in Section SM-7. Column C1 contains the running number of the table entry, columns C2–C3 show the specification and symbol of the quantity of interest; column C4 displays the corresponding table number in the Appendix A; columns C5–C7 and C8–C10 present the sample interval and resolution of pressure and temperature, respectively; columns C11–C12 display the maxima of the relative and absolute deviations, respectively, of the analytically determined quantity *X* from its numerical reference value $X_{\mathrm{num}}$; and columns C13–C14 contain the pressure and temperature at which the maximum deviations occur. As a general tendency, the deviation was found to increase upon decreasing pressure and temperature. The largest relative deviation over all quantities and throughout the analyzed pressure and temperature ranges occurs for the isothermal compressibility $\kappa_{T}$, amounting −2.58‰ at *p*≈0.1MPa and T=235.15K. Starting from the values at the maximum deviation, the deviation rapidly decreases to the parts-per-million range upon increasing pressure and temperature.

In order to assess the applicability of the two-step predictor-corrector scheme proposed here, the deviations reported in Table 1 (misfit of the analytical form of the EoS) are compared with the deviations of experimental data from the values calculated using the IAPWS G12-15 [36] formulation (misprediction of the numerical form of the EoS) as reported in Holten et al. [23]. The analysis delivered the following results.
Mass density:
-Table 1, entry (1) (atmospheric pressure): The maximum relative deviation of the analytically determined mass density from the numerical one amounts to 0.172‰ at *p*=0.101325MPa and *T*=235.15K. Holten et al. [23] (Figure 6 therein) found the experimental density data deviating from the IAPWS G12-15 [36] formulation by ±0.2‰ at *p*=0.101325MPa and *T*≤255K. Hence, for the mass density the misfit of the analytical form of the EoS remains still smaller than the misprediction of the numerical form of the EoS.-Table 1, entry (2) (high pressure): The maximum of the relative deviation of the analytical EoS form from the numerical one amounts to 221ppb at *p*=0.1MPa and *T*=253.15K. Holten et al. [23] (Figure 7 therein) found the experimental density data deviating from the IAPWS G12-15 [36] formulation by more than −0.1% at *T*=253.15K and *p*≥ 200 MPa. The deviation at *p*=0MPa cannot be resolved at the chosen scale (percent). However, one can expect that in the specified pressure and temperature range the misfit of the analytical form of the EoS does not relevantly exceed the misprediction of the numerical form of the EoS.-Table 1, entry (3) (very high pressure): In the specified pressure and temperature range, the maximum of the relative deviation of the analytically determined mass density from the numerical one amounts 0.79ppb at *p*=0.1MPa and T=273.15K. Throughout most of the (T,p) space depicted in Holten et al. [23] (Figure 21 therein) this value is much smaller than the deviations of the experimental density from the IAPWS G12-15 [36] formulation. At atmospheric pressure, the latter, however, cannot be resolved in Holten et al. [23] (Figure 21 therein) at the chosen scale (percent). Thus, also for the very high pressure range one can safely conclude that the misfit of the analytical form of the EoS does not exceed the misprediction of the numerical form of the EoS.Thermal expansivity:
-Table 1, entry (5): The maximum of the deviation of the analytically determined expansivity from the numerical one amounts $0.11\cdot 10^{-6}\;K^{-1}$ at *p*=0.1MPa and *T*=245.5K. At these conditions the deviation of the experimental values from the IAPWS G12-15 [36] formulation reported in Holten et al. [23] (Figure 9 therein) amounts to $-0.2\cdot 10^{-4}\;K^{-1}$, i.e., the misfit of the analytical form of the EoS is much smaller than the misprediction of the numerical form of the EoS (see also Holten et al. [23], Figure 10 therein).Isothermal compressibility:
-Table 1, entry (6): The maximum of the deviation of the analytically determined compressibility from the numerical one amounts to −2.58‰ at *p*=0.101325MPa and T=235.15K. At these conditions the deviation of the experimental values from the IAPWS G12-15 [36] formulation reported in Holten et al. [23] (Figure 15 therein) amounts to ≈6%, i.e., the misfit of the analytical form of the EoS is much smaller than the misprediction of the numerical form of the EoS (see also Holten et al. [23], Figure 16 therein).Isobaric heat capacity:
-Table 1, entry (7): The maximum of the deviation of the analytically determined heat capacity from the numerical one amounts to −2.35‰ at *p*=0.101325MPa and T=235.15K. At these conditions, the deviation of the experimental values from the IAPWS G12-15 [36] formulation reported in Holten et al. [23] (Figure 19 therein) amounts up to −8%, i.e., the misfit of the analytical form of the EoS is much smaller than the misprediction of the numerical form of the EoS.Sound speed:
-Table 1, entry (9) (high pressure): The maximum of the deviation of the analytically determined sound speed from the numerical one over the specified pressure and temperature ranges amounts to 821ppb at *p*=0.1MPa and *T*=253.15K. At these conditions the deviation of the experimental values from the IAPWS G12-15 [36] formulation reported in Holten et al. [23] (Figure 18 therein) amounts up to −0.2%, i.e., the misfit of the analytical form of the EoS is much smaller than the misprediction of the numerical form of the EoS (see also Holten et al. [23], Figure 17 therein).-Table 1, entry (10) (very high pressure): The maximum of the deviation of the analytically determined sound speed from the numerical one over the specified pressure and temperature ranges amounts to 4ppb at *p*=0.1MPa and *T*=273.15K. At these conditions, the deviation of the experimental values from the IAPWS G12-15 [36] formulation reported in Holten et al. [23] (Figure 22 therein) amounts to ≤0.1%, i.e., the misfit of the analytical form of the EoS remains much smaller than the misprediction of the numerical form of the EoS.

For the Gibbs free energy and the entropy (entries (11) and (12)) no experimental data for direct comparison are available.

## 6. Conclusions

Possible implications of the two-state nature of subcooled water for ice crystallization have been discussed. A new two-step predictor-corrector method for the approximative determination of the low-density water fraction of subcooled water has been developed which allows a highly accurate determination of the specific Gibbs energy of subcooled water at given pressure and temperature. The misfit of this analytical solution against the exact numerical solution was demonstrated to be smaller than the misprediction of the original IAPWS G12-15 [36] formulation. The solution proposed here is intended to be used for highly frequent calls of the IAPWS G12-15 [36] formulation in CFD applications. Appendix B contains the algorithm for the calculation of the equilibrium mole fraction of low-density water in subcooled water and the thermodynamic reference values for the check of the correct computer implementation.

The calculus employed here is based on a polynomialization of the underlying transdendental equation (Equations (Equation 6) and (Equation 8)) by expansion of its exponential term into a Taylor series up to the quadratic term (Equation (Equation 9)). This resulted in a cubic equation (Equation (Equation 10)), for which an analytical solution formula exists. According to the Galois theory, the existence of a general solution formula for root finding (employing the four basic arithmetic operations and root extraction only) is restricted to polynomials with a degree of up to four, i.e., quintic and higher-degree equations must be solved numerically. Therefore, if the underlying transcendental equation can be taylorized into a polynomial with a degree of lower than or equal to four one can derive analytical solutions. The approach presented here is restricted to cubic equations, i.e., it can be applied to any other EoS of water (or other fluids) provided this equation can be taylorized into a cubic polynomial. For a quaternary polynomial a more generalized solution formula is available.

## Figures and Tables

**Figure 1 entropy-22-00933-f001:**
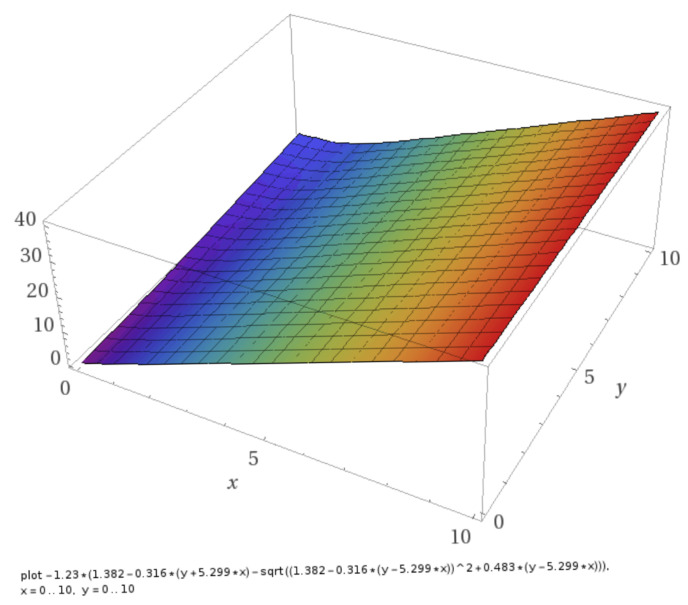
Ordering field L(x = τ,*y* = π) according to Appendix A in the interval 0≤τ≤10 and 0≤π≤10. Graphic plot using Wolfram Alpha LLC, 2020, https://www.wolframalpha.com/examples/mathematics/plotting-and-graphics/ (accessed on 22 July 2020).

**Table 1 entropy-22-00933-t001:** Summary of the evaluation of the tables presented in Appendix A. Column C1 contains the running number of the table entry, columns C2–C3 show the quantity of interest and its corresponding symbol; column C4 displays the corresponding table number in the Appendix A; columns C5–C7 and C8–C10 present the sample interval and the resolution of the pressure and temperature, respectively; columns C11–C12 display the maxima of the relative and absolute deviations, respectively, of the analytically determined quantity *X* from its numerical reference value $X_{\mathrm{num}}$ (misfit); and columns C13–C14 contain the pressure and temperature at which the maximum deviations occur.

C1	C2	C3	C4	C5	C6	C7	C8	C9	C10	C11	C12	C13	C14
Entry	Quantity	Symbol X	Table	Pressure Range			Temperature Range			${\left(\frac{X-X_{\mathrm{num}}}{X_{\mathrm{num}}}\right)}_{\max}$	${\left(\frac{X-X_{\mathrm{num}}}{10^{-4}\;K^{-1}}\right)}_{\max}$	(p,T) at max. dev.	
				$\frac{p_{\min}}{\mathrm{MPa}}$	$\frac{p_{\max}}{\mathrm{MPa}}$	$\frac{\Delta p}{\mathrm{MPa}}$	$\frac{T_{\min}}{K}$	$\frac{T_{\max}}{K}$	$\frac{\Delta T}{K}$	ppb		$\frac{p}{\mathrm{MPa}}$	$\frac{T}{K}$
(1)	Mass density	$\hat{\varrho }$	SM-7.1.1	0.101325			235.15	300.15	5	$0.172\cdot 10^{6}$		0.101325	235.15
(2)			SM-7.1.2	0.1	400	5	253.15	303.15	≈2−5	221		0.1	253.15
(3)			SM-7.1.3	0.1	1000	50	273.15	303.15	≈5−20	0.79		0.1	273.15
(4)			SM-7.1.4	0.02	0.1	0.01	235.15	300.15	5	$0.174\cdot 10^{6}$		0.02	235.15
(5)	Expansivity	$\alpha_{p}$	SM-7.2.1	0.1	600	10	245.5	288.0	5−10		$0.11\cdot 10^{-2}$	0.1	245.5
(6)	Compressibility	$\kappa_{T}$	SM-7.3.1	0.1	190	≈10−50	235.15	300.15	5	$-2.58\cdot 10^{6}$		0.101325	235.15
(7)	Heat capacity	${\hat{c}}_{p}$	SM-7.4.1	0.1	190	≈10−50	235.15	300.15	5	$-2.35\cdot 10^{6}$		0.101325	235.15
(8)			SM-7.4.2	0.02	0.1	0.01	235.15	300.15	5	$-2.37\cdot 10^{6}$		0.02	235.15
(9)	Sound speed	*w*	SM-7.5.1	0.1	400	≈10−50	253.15	303.15	≈2−5	821		0.1	253.15
(10)			SM-7.5.2	0.1	1000	≈50	273.15	303.15	10	4		0.1	273.15
(11)	Gibbs energy	$\hat{g}$	SM-7.6.1	0.02	0.1	0.01	235.15	300.15	5	−955		0.02	235.15
(12)	Entropy	$\hat{s}$	SM-7.6.2	0.02	0.1	0.01	235.15	300.15	5	$-0.86\cdot 10^{6}$		0.02	235.15

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
