# Peer review of "Analytical Determination of the Nucleation-Prone, Low-Density Fraction of Subcooled Water"

_entropy, 2020, doi:10.3390/e22090933_

Round 1
Reviewer 1 Report
This detailed study is carried out in a very good fashion and the two-step predictor-corrector method is clearly presented. The interest in such new methods is mildly high in the research field so that this paper, in my humble opinion, can be accepted after minor spell check.
Eventually, if in need of page suppression (considering the huge size in pages), i advise the authors to suppress Appendix B since the results are of historical interest only.
Typos:
line -- text -- suggested correction
8 -- trancendental equation -- transcendental equation
12 -- quantitites -- quantities
199 -- dimensionsless -- dimensionless
xxx (before eq. (7) ) -- ..subscript ’e’ for the equlibrium -- equilibrium
202 & 204 -- prerequiste -- prerequisite
346 -- preassure -- pressure
393 -- Apppendix -- Appendix
APPENDIX A.1 (after (A.1) eq.) -- The other quantitites -- quantities
Reviewer 2 Report
The paper is devoted to the analytic determination of thermodynamic quantities of supercooled water, including the Gibbs free energy and entropy. The metastable water was considered as a mixture of low-density and high-density liquid fractions, corresponding to Ostwald’s rule. The good agreement between the obtained computed values and the experimental data was found. The theoretical background, the computational procedure, and the obtained numerical results are presented clearly. I would recommend the authors mention the possibility (if valid) of the principal use of their calculation procedure to the determination of the thermodynamic properties of other subcooled liquids. Overall, in my opinion, this manuscript can be published in Entropy after minor revision.
Questions and remarks:
- Abstract, line 12. “all other thermodynamic quantities”. The calculated quantities should be listed.
- Page 1, line 27. The phrase “pressure and temperature dependence of … nucleation” is not clear and should be reformulated.
- Page 2, line 62. “the lower limit below which it is very difficult to subcool water”. This temperature limit depends on the pressure, therefore “at fixed pressure p=1013.25 hPa” should be added.
- Page 4, Eq. 2. In limit case when the subcooled liquid contains only B fraction, i.e. x=1, the Eq. 2 has a singularity due to ln0. This boundary condition should be discussed.
- Page 2, after Eq. 2. “T the temperature” should be replaced by “T is the temperature”.
- Page 4, Eq. 3. The quantity K in the left part of Eq. 3 should be defined.
- Page 6, Eq. 6. The extremes xe=0 and xe=1 should be discussed since they result to the singularity of Eq. 6.
Reviewer 3 Report
In this paper, the authors try to show the improvement achieved in the mathematical methodology used for the determination of the thermodynamic parameters of subcooled water. The authors proposed a new method, in two steps, for the estimation of the low-density subcooled water.
The work done is quite exhaustive, however there are some points that should be solved before the publication of the paper.
First of all, I recommend a structural change in order to follow the structure used in the majority of the scientific journals, that is:
- Introduction
- Materials and methods (experimental and numerical setup)
- Simulation experiments (theoretical background)
- Application to case study
- Discussion
- Conclusion
On the other hand, the use of abbreviations (or not) has to be consistent along the text (as an example, line 53: w.r.t. (with regard to) while in other parts is full written).
The use of footnotes and textual quotations is quite odd in scientific papers. It is common in PhD dissertations, but it is not for papers. Therefore, the authors have to rewrite to include in the text what they consider important. Of course, if English is the language of the paper, all the text has to be in English (for example, Footnote (1) (2) (3): if it is important for the paper, should be translated to English and introduced into the text. If not, delete). Regarding the quotations, please, the authors have to summarized and re-write using their own words (see lines 119-122; 134-138).
It is difficult to find the most important part of the paper: the improvement achieved with the new method proposed. My recommendation would be shorten the explanations and focus on the mathematical method applied. Maybe some tables with the changes and differences between the new method and the old one, could be interesting.
For all those reason my recommendation is a MAJOR REVISION.
Author Response
The response to referee's 1, 2, and 3 is attached below. Please note the changes in our updated reply to 1 and 2, originating from the major revision demanded by referee 3 (received after our response to 1 and 2).
Many thanks for the great work done by the three referees!
Olaf Hellmuth

Round 2
Reviewer 1 Report
After the major revision it appears to me that now the paper can safely be published. I appreciate the intense revision work done from the authors after the suggested corrections!
Reviewer 2 Report
I have read the authors' responses to the comments. The authors answered all my questions and made the corresponding changes in the paper. In my opinion, the revised manuscript has improved significantly and can be published in Entropy.